# Small signal analysis for the characterization of organic electrochemical transistors

Youngseok Kim ®[1] ✉, Joost Kimpel[1], Alexander Giovannitti[1] &
Christian Müller ®[1] ✉

A method for the characterization of organic electrochemical transistors (OECTs) based on small signal analysis is presented that allows to determine the electronic mobility as a function of continuous gate potential using a standard two-channel AC potentiostat. Vector analysis in the frequency domain allows to exclude parasitic components in both ionic and electronic conduction regardless of film thickness, thus resulting in a standard deviation as low as 4%. Besides the electronic mobility, small signal analysis of OECTs also provides information about a wide range of other parameters including the conductance, transconductance, conductivity and volumetric capacitance through a single measurement. General applicability of small signal analysis is demonstrated by characterizing devices based on n-type, p-type, and ambipolar materials operating in accumulation or depletion modes. Accurate benchmarking of organic mixed ionic-electronic conductors through small signal analysis can be anticipated to guide both materials development and the design of bioelectronic devices.

Organic electrochemical transistors (OECTs) receive considerable interest as sensor devices and as a basic building block of more advanced bioelectronic circuitry[1–5]. Moreover, OECTs are widely employed for the characterization of organic mixed ionic-electronic conductors (OMIECs), which constitute the channel material. Hence, accurate determination of the device performance[6–8] as well as in-depth studies of the electrochemical properties of OMIEC materials[9–12] require accurate characterization methods. The most important parameters are the volumetric capacitance $C^*$, representing the change of the number of charge carriers stored per unit volume upon a small fluctuation in potential, and the mobility $\mu$, i.e. the electric field-normalized velocity of electronic charge carriers[13]. The product $[\mu C^*]$ is often used to benchmark OMIEC materials, but techniques are lacking that allow to determine the two parameters and in particular $\mu$ independently, which leads to ambiguities when comparing materials[14].

A widely used method for the determination of $C^*$ and $\mu$ involves two measurements: (1) OECT characterization and the analysis of transfer curves and (2) electrochemical impedance spectroscopy (EIS)[13]. OECT measurements allow to determine the volumetric transconductance $g_m^*$

from transfer curves, i.e. the volumetric source-drain current $I_{DS}^*$ is recorded as a function of gate potential $V_{GS}$, according to:

$$g_m^* = g_m / \frac{wd}{L_{ch}} = \frac{dI_{DS}^*}{dV_{GS}} = \begin{cases} -[\mu C^*] \cdot V_{DS}, & \text{linear regime} \\ [\mu C^*] \cdot (V_{GS} - V_{th}), & \text{saturation regime} \end{cases} \quad (1)$$

where $g_m$ is the transconductance, $w$, $d$ and $L_{ch}$ are the width, thickness, and length of the channel, respectively, $V_{DS}$ is the drain potential and $V_{th}$ is the threshold voltage. The product $[\mu C^*]$ can thus be obtained from transfer curves in either the linear ($V_{DS} > V_{GS} - V_{th}$ for p-type and $V_{DS} < V_{GS} - V_{th}$ for n-type OECTs) or saturation regime ($V_{DS} < V_{GS} - V_{th}$ for p-type and $V_{DS} > V_{GS} - V_{th}$ for n-type OECTs)[9]. Subsequently, $\mu$ is obtained by dividing $[\mu C^*]$ by corresponding $C^*$ values, which must be independently determined through EIS (see Table 1). The use of two distinct characterization methods for the determination of $[\mu C^*]$ and $C^*$ introduces uncertainty in the calculated mobility due to the propagation of errors. In addition, the OECT active layer, which swells during operation due to the uptake of hydrated ions (i.e., ions that are accompanied by water molecules[15]), is typically

[1]Department of Chemistry and Chemical Engineering, Chalmers University of Technology, Göteborg, Sweden. ✉e-mail: ykim@chalmers.se; christian.muller@chalmers.se

only tens of nanometers thin. As a result, a large relative error in $d$ can arise due to an uneven film thickness across the channel, the adventitious uptake of water, as well as errors inherent to film thickness measurements with atomic force microscopy (AFM) or surface profilometry. An analysis of the recent literature indicates a coefficient of variation $\hat{c}_v = SD/\bar{\mu}$ of up to 100% for mobility values obtained via the traditional transfer curve/EIS method (standard deviation $SD$ and mean mobility $\bar{\mu}$; Fig. 1).

An alternative method for the determination of $\mu$ is an analysis of the transient response in either the time or frequency domain. The transit time $\tau_e$ of electronic charge carriers through the channel is obtained by comparing the current values at the gate and drain electrodes, $I_{GS}$ and $I_{DS}$, in case of a constant gate current[16] or a sinusoidal gate potential signal[17] according to:

$$\frac{dI_{DS}}{dt} = -I_{GS}/\tau_e \qquad (2)$$

Then, $\mu$ can be obtained regardless of the active-layer thickness according to:

$$\mu = \frac{L_{ch}^2}{\tau_e V_{DS}} \qquad (3)$$

**Table 1 | Input/output signals and extractable parameters G, $\sigma$, $C^*$, $g_m$, $[\mu C^*]$ and $\mu$ for conventional OECT characterization methods (transfer curve/EIS, constant gate current, sinusoidal gate potential) and small signal analysis**

|  | Transfer curve | EIS | Constant gate current | Sinusoidal gate potential | Small signal analysis |
|---|---|---|---|---|---|
| Input signal | $V_{GS}, V_{DS}$ | Stepwise $V_{DC}$ + $V_{AC}$ | Stepwise $I_{GS}, V_{DS}$ | Stepwise $V_{DC}$ + $V_{GS,AC}, V_{DS}$ | Continuous $V_{GS,DC}$ + $V_{GS,AC}, V_{DS}$ |
| Output signal | $I_{GS}, I_{DS}$ | $I_{AC}$ | $V_{GS}, I_{DS}$ | $I_{GS,AC}, I_{DS,AC}$ | $I_{GS,DC}, I_{GS,AC}, I_{DS,DC}, I_{DS,AC}$ |
| $G, \sigma, g_m$ | ✓ | – | ✓ | ✓ | ✓ |
| $[\mu C^*]$ | ✓ | – | ✓ | ✓ | ✓ |
| $C^*$ | – | ✓ | ✓ | ✓ | ✓ |
| $\mu$ | (✓) | – | ✓ | ✓ | ✓ |

While this approach facilitates a more accurate determination of $\mu$ than the method using transfer curves and EIS, repeated measurements with a stepwise gate current or gate potential would need to be carried out to obtain information of the change in $\mu$ as a function of gate potential. Further, the extracted $\mu$ tends to strongly deviate from the true value once the electrochemical redox peak dominates the output signal. It should be noted that $I_{GS}$ comprises both the non-Faradaic (capacitive) current as well as the Faradaic (resistive) current. Faradaic processes, which could be identified via the detection of hydrogen or the oxygen reduction response, would need to be ruled out for precise characterization (see Table 1 for summary of characterization methods)[18,19].

Here, we introduce a method for the determination of $\mu$ that is based on vector analysis in the frequency domain and small signal analysis. For OECTs with a well-defined active layer, the mobility can be reliably extracted regardless of the channel thickness, yielding a very low $\hat{c}_v = 4\%$ for 40 devices (Fig. 1). This is achieved by counting and comparing the number of charge carriers associated with the non-Faradaic part of $I_{GS}$ and the corresponding $I_{DS}$. The here developed method uses a mixed $V_{GS}$ signal composed of a linear sweep to which a small sinusoidal signal is added. This approach allows to simultaneously characterize devices under steady-state and transient conditions, yielding device parameters as a function of continuous gate potential. Thus, essential parameters such as the conductance $G$, $g_m^*$, the conductivity $\sigma$, $C^*$ and $\mu$ can be extracted through one single measurement (Table 1), which can be readily carried out with a commercial two-channel AC potentiostat. Finally, versatility of the developed small-signal analysis method is demonstrated by characterizing OECTs based on a range of n-/p-type and ambipolar materials as well as accumulation/depletion mode materials.

## Results and discussion

We prepared OECTs with a $L_{ch} = 20\,\mu m$ and $w = 100\,\mu m$ with two contact regions with a length $L_{contact} = 100\,\mu m$ that served as the source and drain electrode (Fig. 2a, see method section for fabrication procedure). A three-electrode configuration was used consisting of an Ag/AgCl reference and a Pt counter electrode. $V_{GS}$ comprised two waveforms, i.e. a pseudo steady-state triangular potential ($V_{GS,DC}$, from +0.4 to −0.6 V, scan rate $SR = 10\,mV\,s^{-1}$) and a sinusoidal AC small potential ($V_{GS,AC} = A\sin(2\pi f_{AC}t)$, amplitude $A = 10\,mV$, frequency $f_{AC} = 10\,Hz$), while a constant $V_{DS} = 10\,mV$ was applied. The conjugated polymer p($g_3$TT-T2) with triethylene glycol side chains was used as the

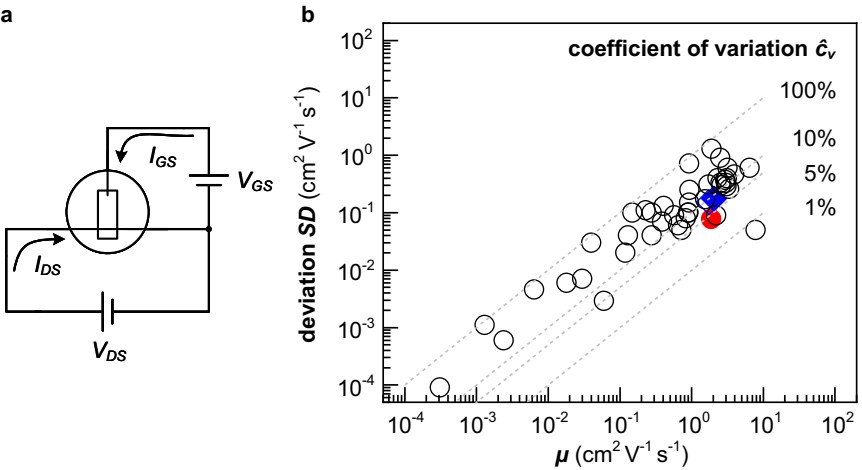

**Fig. 1 | Correlation between the mean mobility $\bar{\mu}$ and the coefficient of variation $\hat{c}_v$. a** Measurement scheme for the characterization of OECT devices. **b** Coefficient of variation $\hat{c}_v = SD/\bar{\mu}$, where $\bar{\mu}$ and $SD$ are the mean mobility and corresponding standard deviation, extracted from literature since 2014 (open symbols)[6,9,10,31–46] and obtained in this work using the conventional transfer curve/EIS method (blue diamond) and small signal analysis (red circle); dashed lines represent $\hat{c}_v = 1, 5, 10$ and 100%.

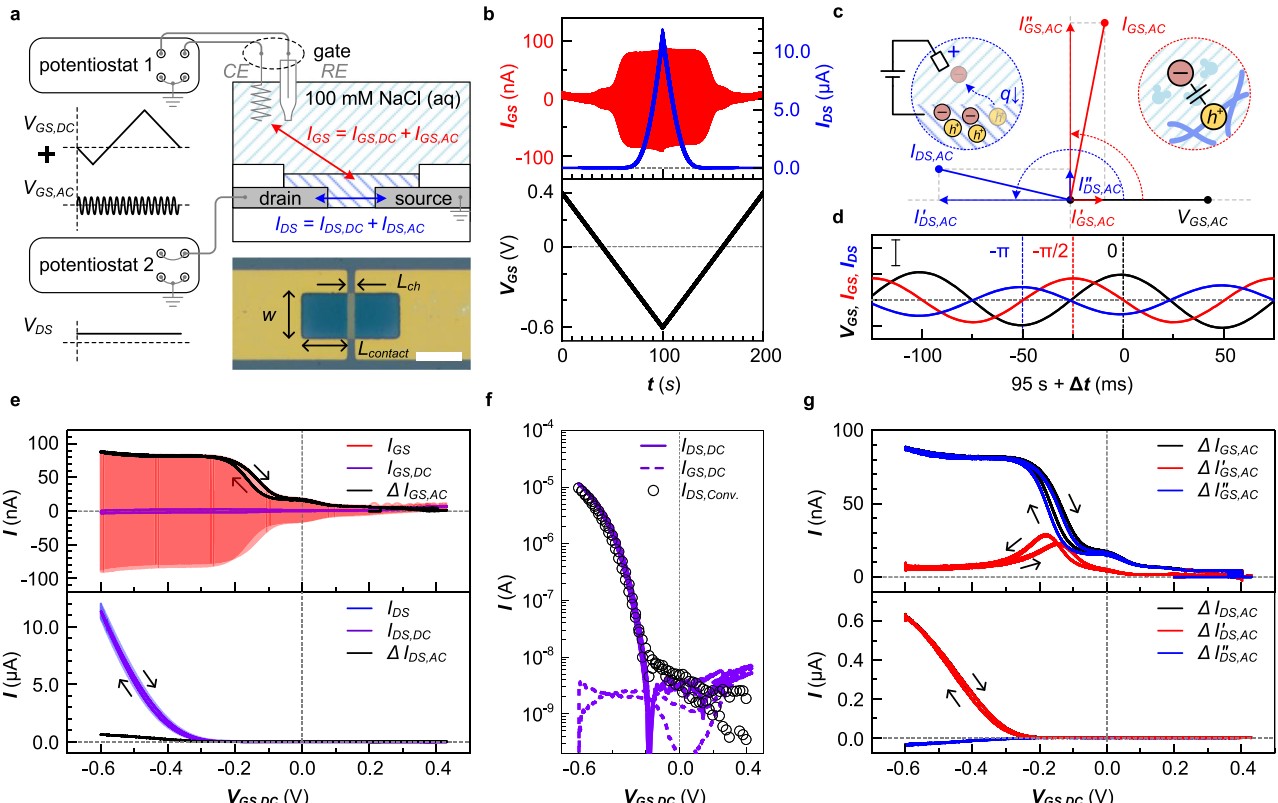

**Fig. 2 | Small signal analysis for OECT device characterization. a** Device and measurement scheme for small signal analysis. A gate potential $V_{GS}$ consisting of a triangular potential $V_{GS,DC}$ and a small-amplitude sinusoidal potential $V_{GS,AC}$ is applied to the 100 mM NaCl aqueous electrolyte via a three-electrode configuration with a counter ($CE$) and a reference electrodes ($RE$), with a constant drain potential $V_{DS}$. The lower inset depicts an optical microscopy image of the channel region of an OECT device (scale bar = 100 μm). **b** Time-varying input gate potential $V_{GS}$ (black) as well as output gate $I_{GS}$ (red) and drain current $I_{DS}$ (blue). **c** Phasor diagram of output currents relative to $V_{GS}$, and (**d**) traces of $V_{GS}$ (black), $I_{GS}$ (red) and $I_{DS}$ (blue) at $t = 95\,s + \triangle t$, and centered at $V_{GS} = 0.55\,V$, $I_{GS} = 0\,A$, and $I_{DS} = 8\,μA$ (y-axis scale bar = 10 mV, 100 nA, and 1 μA for $V_{GS}$, $I_{GS}$, and $I_{DS}$). Since the holes are

accumulated through the electrical double layer in case of p-type accumulation mode OECTs (see inset of **c**), the gate and drain currents show a time delay from the gate input bias of π/2 and π, respectively. **e** Amplitude of AC and DC components of $I_{GS}$ (upper panel) and $I_{DS}$ (lower panel) as a function of $V_{GS,DC}$. **f** Steady-state current response $I_{DS,DC}$ (solid line) and $I_{GS,DC}$ (dashed line) as a function of the $V_{GS,DC}$ from small signal analysis, and transfer curve $I_{DS}$ vs. $V_{GS,DC}$ from conventional analysis (open circles). **g** Amplitude of $I_{GS,AC}$ (upper panel) and $I_{DS,AC}$ (lower panel) and its real and imaginary components as a function of $V_{GS,DC}$, obtained from small signal analysis. The real (red) and imaginary component (blue) were extracted from the output current values (black) by vector analysis as shown in **c**.

active material (see Supplementary Fig. S1 for chemical structure), which features a high electronic mobility upon electrochemical oxidation (Supplementary Figs. S2 and S3) with good stability during cyclic operation[20].

OECT devices with a $V_{DS} = 10\,mV$ were operating in the linear regime within the potential window of $V_{GS}$ ranging from −0.6 V to +0.4 V (Supplementary Fig. S3b). Thus, $I_{GS}$ can be described by the Bernards–Malliaras model[16]:

$$I_{DS} = [\mu C^*]\frac{wd}{L}\left((V_{GS} - V_{th})V_{DS} - \frac{1}{2}V_{DS}^2\right) \qquad (4)$$

We recorded $I_{GS}$ and $I_{DS}$ as a function of time $t$, which are the mixed current responses from the steady-state $V_{GS,DC}$ and sinusoidal $V_{GS,AC}$ input signals. As $V_{GS,DC}$ is scanned from +0.4 to −0.6 V, the amplitude of $I_{GS}$ increases and reaches a plateau, while $I_{DS}$ gradually increases. Considering the electrochemical double layer at the interface between the electrolyte and active layer[21] and the p-type operation of the material[20], a sinusoidal current response at the gate and drain electrode, $I_{GS,AC}$ and $I_{DS,AC}$ is expected with a phase shift of −90° and −180° relative to the $V_{GS,AC}$ signal, respectively. Hence, we can rule out any unintended redox response (e.g., oxygen reduction response[22]), which would contribute an additional real component to $I_{GS}$ and a corresponding imaginary component to $I_{DS}$, resulting in an additional

phase shift. We also separated the measured $I_{GS}$ and $I_{DS}$ values into the real ($I'_{GS,AC}$ and $I'_{DS,AC}$) and imaginary components ($I''_{GS,AC}$ and $I''_{DS,AC}$; Fig. 2c). The measured current traces at the gate and drain electrodes show an expected phase shift near −90° and −180° from the signal of the gate potential at $t = 95\,s$ when the active layer is heavily doped (Fig. 2d).

Subsequently, Fourier transform analysis was used to separate the current responses at the gate and drain electrodes from the mixed gate input potential (Fig. 2e). The resulting steady-state gate and drain current response ($I_{GS,DC}$ and $I_{DS,DC}$) as a function of $V_{GS,DC}$ are in very good agreement with the transfer curves recorded in the linear regime, which were obtained through the conventional characterization method (Fig. 2f; see Method section for details). Upon dedoping of the active material through application of a positive $V_{GS,DC}$, the amplitude of $I'_{GS,AC}$ (denoted as $\Delta I'_{GS,AC}$) approaches zero, while the amplitude of $I''_{GS,AC}$ ($\Delta I''_{GS,AC}$) has a value of 4.5 nA. This offset is explained with the capacitance of the electrical double layer between the active layer and the underlying metal electrode. As the $V_{GS,DC}$ is changed to negative values, $\Delta I''_{GS,AC}$ gradually increases and reaches a plateau with 85 nA, reflecting the electrochemical capacitance $C$ of the active material (*vide infra*). At $V_{GS,DC} = -0.15\,V$, a distinct peak in $\Delta I'_{GS,AC}$ is observed in both backward and forward scans (Fig. 2g). This peak is assigned to the contact resistance between the active layer and metal electrode (Supplementary Fig. S2d), which is also observed in corresponding EIS

measurements (Supplementary Fig. S4; *c.f.* the resistive response showing a flat region near $f_{AC}$ 10 Hz in the Bode plot). Unlike the EIS measurement, the small signal analysis is conducted with one frequency value, so that the parasitic response is superimposed onto the current value.

We used the output current values obtained from the small signal analysis to extract material and device parameters as a function of $V_{GS,DC}$. Initially, the conductance $G$ of the active material was determined by using a steady-state input $V_{DS,DC}$, which resulted in an output $I_{DS,DC}$ with $G = I_{DS,DC}/V_{DS,DC}$ (Fig. 3a). The conductivity $\sigma$ was obtained by normalizing $G$ with $wd/L_{ch}$ (Fig. 3b). Concurrently, the capacitance $C$, *i.e.* the accumulation of additional charge carriers $dq$ upon a change in electrochemical potential $dV_{GS}$, was extracted according to the ionic non-Faradaic current value ($I''_{GS,AC} = dq/dt$). Then, $C^*$ can be obtained according to (see Supplementary Information for derivation of Eq. (5)):

$$C^* = \frac{\Delta I''_{GS,AC}}{2\pi f_{AC} \, \Delta V_{GS,AC} \cdot vol} \tag{5}$$

where, $vol = wd \cdot (L_{ch} + 2L_{contact})$ is the volume of the active layer (Fig. 3c). The change in $C^*$ as a function of $V_{GS,DC}$ indicates that the active layer material is gradually doped for $V_{GS,DC} < -0.1$ V, which agrees with the onset potential extracted from the x-axis intercept of a linear fit near the maximum slope of the current transient of p(g₃TT-T2) recorded during oxidation ($E_{onset} = +0.1$ V, vs. Ag/AgCl, see Supplementary Fig. S2a). We would like to point out that $C^*$ obtained from the small signal analysis tends to be underestimated compared with values obtained from EIS characterization (Supplementary Fig. S2d), which can be explained with limited ionic motion at $f_{AC} = 10$ Hz. A decrease in $f_{AC}$ would enable a more accurate measurement of $C^*$ (see also Supplementary Fig. S5d).

The transconductance $g_m$ was extracted from the steady-state ($g_{m,DC} = \partial I_{DS,DC}/\partial V_{GS,DC}$) as well as transient response ($g_{m,AC} = \partial I'_{DS,AC}/\partial V_{GS,AC}$). The benchmarking parameter $[\mu C^*]$ was obtained from both the DC and AC response via Eq. (1), denoted as $[\mu C^*]_{DC}$ and $[\mu C^*]_{AC}$, respectively (Fig. 3d, e). Note that $[\mu C^*]_{AC}$ is identical to the product of $\mu_{AC}$ (*vide infra*) and $C^*$, which were extracted separately from the transient response. Since small signal analysis underestimates $C^*$, values for $[\mu C^*]_{AC}$ obtained from the AC response are about 7% smaller than $[\mu C^*]_{DC}$ from the DC response. Before extracting the mobility, the transit time $\tau_e$, the time taken for charge carriers to traverse the channel, was obtained according to Eq. (2), with $I_{GS}$ given by the non-Faradaic gate current, i.e. $I_{GS} = I''_{GS,AC} \cdot A_{ch}/A_{active}$, and $I_{DS}$ given by the modulated drain current, i.e. $I_{DS} = I'_{DS,AC}$. The area correction factor $A_{ch}/A_{active}$, where $A_{ch}$ and $A_{active}$ are the area of the channel region and total active layer, respectively, is introduced to account for the additional charge carriers that are generated in the active layer in contact with the metal electrodes. Then, $\mu_{AC}$ can then be obtained according to Eq. (3). It is noteworthy that the mobility extracted through small-signal analysis does not depend on the thickness $d$ of the active layer. For p(g₃TT-T2), $\mu_{AC}$ gradually increased with increasing doping level for $V_{GS,DC} < -0.3$ V (solid line, Fig. 3f), and a highest value of 1.95 cm² V⁻¹ s⁻¹ was achieved at $V_{GS,DC} = -0.6$ V. At $V_{GS,DC} < -0.6$ V, $\mu_{AC}$ tends to decrease slightly (data not shown), which is expected due to formation of less-mobile bipolarons[23,24] and/or an increased distance between polymer chains due to excessive swelling[25]. Also, the traces agree with those obtained from the steady-state response ($\mu_{DC} = [\mu C^*]_{DC}/C^*$). Moreover, $\mu_{AC}$ only changed by 1% when increasing $f_{AC}$ from 5 to 60 Hz even though $C^*$ decreased by 10% when increasing $f_{AC}$ from 5 to 60 Hz (Supplementary Fig. S5).

As with traditional small signal analysis, which is often used for the analysis of field-effect transistors[26], a decrease in the magnitude of the offset potential between source and drain electrodes i.e., $V_{DS}$, increases the accuracy of the extracted parameters. For the same reason, the slope of $\Delta I_{GS,AC}$ and $C^*$ near the onset potential $V_{GS,DC} = -0.1$ V became

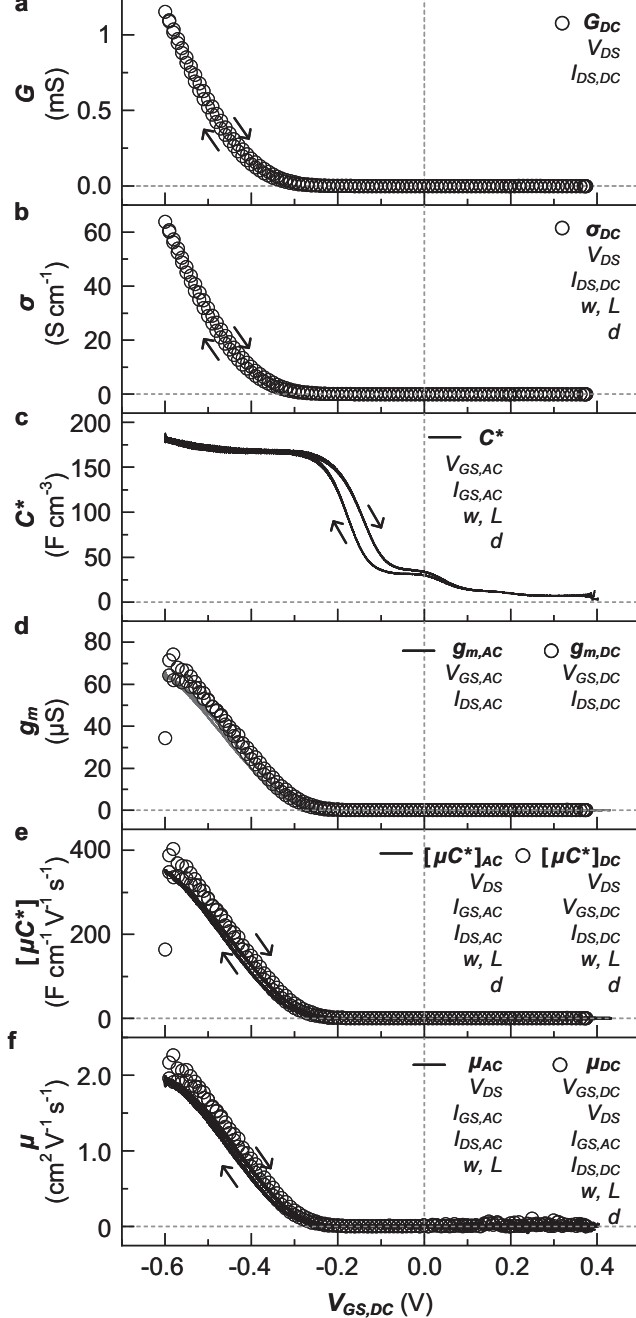

**Fig. 3 | Parameters extracted via small signal analysis.** Traces of (**a**) conductance $G$, (**b**) conductivity $\sigma$, (**c**) volumetric capacitance $C^*$, (**d**) transconductance $g_m$, (**e**) product of mobility and volumetric capacitance $[\mu C^*]$, and (**f**) mobility $\mu$ as a function of continuous $V_{GS,DC}$. The parameters obtained from the transient and steady-state response are depicted with solid lines and open circles, respectively. The required parameters for each calculation are denoted in each panel.

sharper as the amplitude of $V_{DS}$ was decreased from 100 to 1 mV (Supplementary Fig. S6). However, a reduction in $V_{DS}$ also reduces $I_{DS}$ leading to a lower signal-to-noise ratio at $V_{DS} < 5$ mV. Further, a lower scan rate of $V_{GS,DC}$ is desirable not only for minimizing the undesired transient response from the $V_{GS,DC}$ signal, but also for reducing computational errors during Fourier transform analysis (Supplementary Fig. S7).

To compare the accuracy of different OECT characterization methods, we measured 40 devices with each method and compared the extracted $[\mu C^*]$ and $\mu$ values (Fig. 4). The evolution of $[\mu C^*]_{AC}$ with

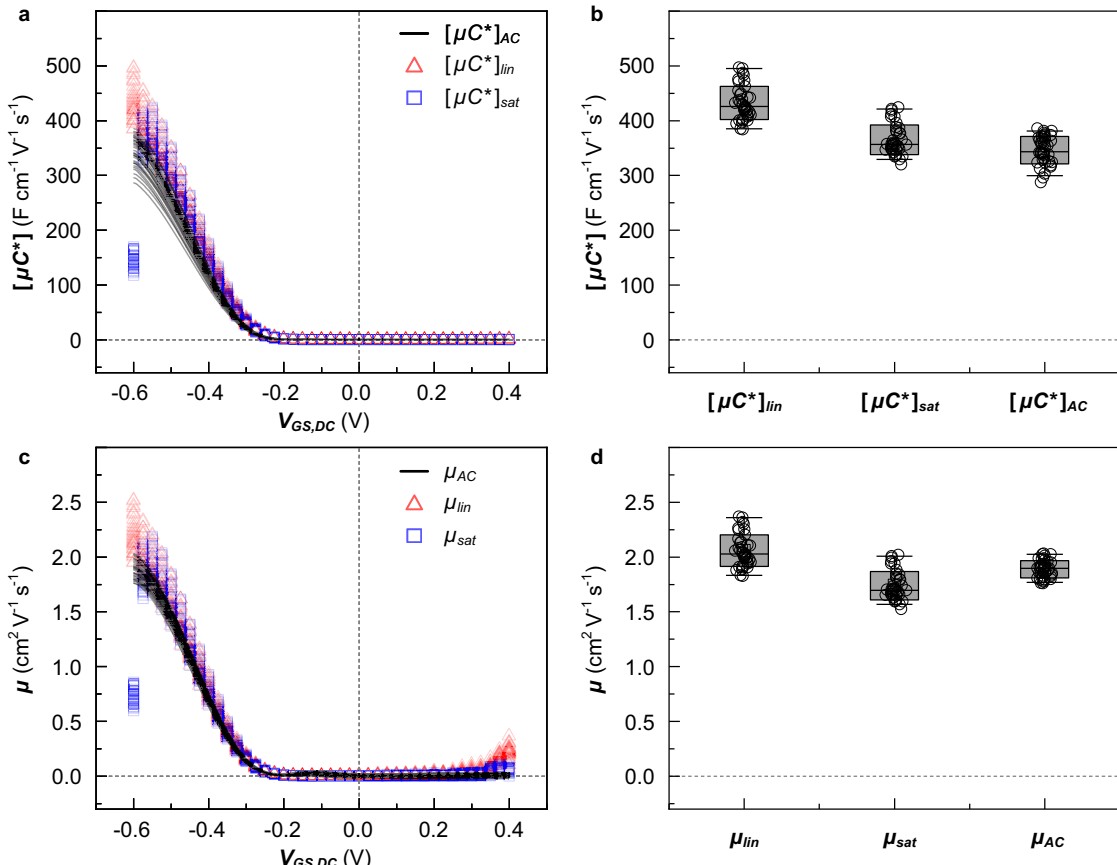

**Fig. 4 | Statistical analysis of the extracted [$\mu C^*$] and $\mu$ values. a, c** [$\mu C^*$] and $\mu$ as a function of $V_{GS,DC}$, measured by small signal analysis (black solid line), and conventional methods from the transfer curves in the linear (red triangles) and saturation region (blue squares), respectively. **b, d** Maximum [$\mu C^*$] and $\mu$ values from characterization of 40 OECT devices (open circles), as well as corresponding mean values and standard deviations (gray boxes).

$V_{GS,DC}$, obtained from small signal analysis, matches that of [$\mu C^*$]$_{lin}$ and [$\mu C^*$]$_{sat}$ obtained through the conventional characterization methods that utilize the transfer curves in the linear or saturation regime, respectively (Fig. 4a). In case of the analysis of transfer curves in the saturation regime, the potential at maximum [$\mu C^*$]$_{sat}$ tends to be shifted from $V_{GS,DC} < -0.6$ V to $-0.55$ V because the operation regime is at the boundary between the saturation and linear regimes (see Supplementary Fig. S3b). The small signal analysis at $f_{AC} = 10$ Hz yields lower [$\mu C^*$] values because $C^*$ is underestimated but that does not affect the extracted value of $\mu$, which is determined independently (see Figs. 3c, 4b and Supplementary Fig. S2d). With regard to the extraction of [$\mu C^*$] values, all three methods require knowledge of the thickness ($d = 36.1 \pm 1.7$ nm from AFM), which results in a similar $SD \approx 7\%$ for [$\mu C^*$]$_{AC}$, [$\mu C^*$]$_{lin}$ and [$\mu C^*$]$_{sat}$. However, the mobility obtained through the small signal analysis has a lower $SD = 4\%$ than values from the conventional methods with $SD \approx 9\%$ (Fig. 4c, d) because only the latter two require knowledge of $d$ as well as $C^*$.

Similar to other characterization methods (see Table 1), the film quality of the active layer, e.g. its roughness and thickness uniformity, is important for an accurate determination of $\mu$ with small signal analysis. Furthermore, it is necessary to consider additional capacitive or resistive components that contribute to ionic conduction such as (1) a parasitic capacitance at the gate electrode (especially in case of a two-electrode configuration with a small gate electrode consisting of a noble metal or OMIEC such as PEDOT:PSS)[27] and/or at the metal source/drain electrode (e.g., electrical double layer capacitance), or (2) a parasitic resistance at the lead wires and interface between the active layer and source/drain metal electrode[28,29]. Those parasitic components can lead to an incorrect gate potential at the channel, an

overestimated $C^*$, and an incorrect drain potential through the channel, respectively.

To establish to which extent small signal analysis can be applied to a wide range of materials, we characterized OECT devices based on various OMIECs including p(g$_4$2T-T) (p-type, accumulation mode), PEDOT:PSS (p-type, depletion mode), and p(gNDI-gT2) (n-type/p-type, accumulation mode) (see Supplementary Note 1 and Supplementary Figs. S8–13 for detailed results of various OMIEC material based OECTs).

The $\mu_{AC}$ and $C^*$ values, which were obtained by the small signal analysis, are summarized in a $\mu - C^*$ plot with four quadrants according to the type of electronic (y-axis) and ionic charge carriers (x-axis) that are associated with the different operation regimes of the various investigated devices (Fig. 5; see Supplementary Note 1 and Supplementary Figs. S8–13 for detailed results of p(g$_4$2T-T), PEDOT:PSS and p(gNDI-gT2) based OECTs). Evidently, the suggested method is valid not only for the evaluation of OECTs based on high-mobility polymers such as p(g$_3$TT-T2) that show hole-anion conduction (p-type, accumulation mode), but also for various devices exhibiting different combinations of hole-cation (p-type, depletion mode), and electron-cation conduction (n-type, accumulation mode). In addition to providing accurate values of the electronic mobility as a function of continuous gate potential, small signal analysis allows to monitor the ionic and electronic behavior simultaneously. As a result, it would allow not only to characterize ionic, electronic and mixed conduction through one single measurement, but also to gain additional information about device degradation during cyclic operation, e.g. by monitoring the charge-carrier density or contact resistance.

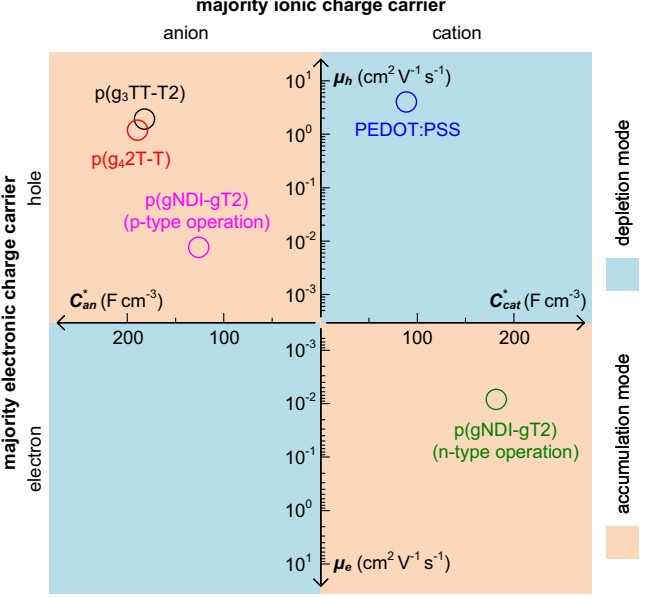

**Fig. 5 | $\mu$ – $C^*$ map with its quadrants representing different combinations of the type of majority ionic/electronic charge carriers within the active layer material.** Orange and blue regions depict accumulation and depletion mode operation, respectively. $C^*_{an}$ and $C^*_{cat}$: volumetric capacitance enabled by the ingression anions and cations, respectively. $\mu_e$ and $\mu_h$: hole and electron mobility, respectively.

We have introduced an accurate method for determination of the electronic mobility of OMIECs, which is based on small signal analysis of OECTs. Gate and drain currents, which were acquired from a commercial two-channel AC potentiostat, were analyzed through vector analysis in the frequency domain, so that parasitic components from the materials and devices could be separated and excluded when calculating various device parameters. The number of charge carriers through the gate and drain electrodes were recorded simultaneously. As a result, the precise determination of the electronic mobility with a low standard deviation of 4% was achieved within a designated gate potential window for 40 OECT devices, which did not require knowledge of the active layer thickness. Moreover, various material and device parameters such as the conductance, transconductance, conductivity and volumetric capacitance could be obtained as a function of the continuous gate potential through one single OECT device measurement. Lastly, the small signal analysis of OECTs could be used for characterizing a range of devices based on both n- and p-type as well as ambipolar materials operating in accumulation or depletion mode, showing that the here introduced method is of general applicability. Small signal analysis allows to accurately benchmark the electronic charge-carrier mobility of OMIECs and likely also other types of materials such as MXenes and semiconducting single-walled carbon nanotubes. It can be anticipated that small signal analysis will aid the identification of best-performing materials that are required for the development of viable bioelectronic devices.

## Methods
### Chemicals and materials
P(g$_3$TT-T2) (number-average molecular weight $M_{n,SEC}$ = 29 kg mol$^{-1}$, $M_{n,NMR}$ = 39 kg mol$^{-1}$, polydispersity index PDI = 2.2) and p(g$_4$2T-T) ($M_{n,SEC}$ = 24 kg mol$^{-1}$, PDI = 3.3) were prepared as previously described[20,30]. p(gNDI-gT2) ($M_{n,SEC}$ = 30 kg mol$^{-1}$, PDI = 2.5) and PEDOT:PSS dispersion (PH-1000) were purchased from 1-Material Inc and Heraeus, respectively. Ethylene glycol (extra pure grade), chloroform (analytical reagent grade) and sodium chloride (analytical reagent grade) were purchased from Thermo Fisher Scientific.

### Organic electrochemical transistor (OECT) device fabrication
Source and drain metal electrodes were defined via a conventional lift-off process using a Karl Suss MA6 contact aligner and a Kurt J Lesker PVD e-beam evaporator on cleaned Marienfeld soda lime glass slides, resulting in channels with a length $L_{ch}$ = 20 μm. Two parylene films were sequentially deposited with a thickness of 1 and 2 μm with an anti-adhesive soap layer between them. Two parylene films were patterned via a conventional dry-etching process using a Karl Suss MA6 contact aligner and reactive ion etcher (O$_2$, 300 W). Then, a solution of the active layer material (7–8 g L$^{-1}$ in chloroform for p(g$_3$TT-T2), p(g$_4$2T-T) and p(gNDI-gT2), and as received PEDOT:PSS solution with 5% ethylene glycol) was spin-coated onto the patterned substrate, followed by peeling away of the second parylene film to pattern the active layer. For PEDOT:PSS thin films, the devices were treated with oxygen plasma before spin-coating for better wetting.

### Electrochemical characterization
Cyclic voltammetry (CV) and electrochemical impedance spectroscopy (EIS) were conducted using 100 mM NaCl aqueous electrolyte and a three-electrode configuration (Ag/AgCl reference electrode (3 M KCl), and Pt wire as the counter electrode) using an electrochemical workstation (Biologic, SP-300). Before and during characterization, the electrolyte was purged with nitrogen gas.

### OECT device characterization
All OECT characterization was conducted with a nitrogen-purged 100 mM NaCl aqueous electrolyte using a three-electrode configuration with an Ag/AgCl reference electrode and Pt counter electrode for applying the gate potential. All analysis was conducted using MATLAB and Origin software (see Data Availability for MATLAB source code).

Steady-state transfer/output type characterization was conducted with two Matlab-controlled source-measure units (Keithley 2400). For the gate electrode the built-in 'four-wire mode' function of the source-measure unit was used. The Ag/AgCl reference electrode and Pt counter electrode, which were immersed in the electrolyte, were electrically connected to the HP and HC ports of the source-measure unit, respectively, and the other LP and LC were connected to the source electrode (HP: high potential, HC: high current, LP, low potential, LC: low current). For the drain-source potential, drain and source electrodes were connected to the high and low ports of the other source-measure unit using a conventional 'two-wire mode'. The applied potential values for transfer and output characteristic curves are described in the manuscript.

Small signal analysis was conducted with a two-channel electrochemical workstation (Biologic, SP-300). Both channels for the gate and drain potential were set to the 'CE to ground' electrode connection mode available through the EC-Lab software, to assign the voltage based on the potential of the source electrode. The former and latter electrode were operated with a three-electrode and two-electrode configuration, respectively. For application of the gate potential, the Ag/AgCl electrode and Pt electrode were connected to the S1 and P1 ports respectively, and the source electrode of the device was connected to the S2, S3 and GND ports (S1: high potential, P1: high current, S2 and S3: low potential, GND: ground). For application of the drain potential, the drain electrode was connected to the P1 and S1 ports, and the source electrode was connected to the S2, S3 and GND electrode ports of the instrument. The gate and drain potentials were assigned with the 'AC-Voltammetry' and 'Constant-Voltage' functions available through the EC-Lab software, respectively (AC-Voltammetry; triangular potential with superimposed small sinusoidal waveform, Constant-Voltage; constant voltage bias, Waveform parameters are described in the manuscript). The specific potential values for each type of device are described in the manuscript. For a synchronized measurement on both channels, the built-in 'synchronize' function was used.

## Data availability
The data used in this study are available in the Zenodo database under accession code 11093663.

## Code availability
The Matlab code for carrying out small signal analysis of OECTs is available in the Zenodo database under accession code 11093663.

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

## Acknowledgements

We gratefully acknowledge financial support from the European Union's Horizon 2020 research and innovation programme through the Marie Skłodowska-Curie grant agreement no. 955837 (HORATES; C.M.), the Knut and Alice Wallenberg Foundation (grant agreement nos. 2021.0295 and 2022.0034; C.M.) and the European Research Council (ERC) under grant agreement nos. 101043417 (C.M.) and 101116071 (A.G.). Myfab is acknowledged for support and access to the nanofabrication laboratory at Chalmers.

## Author contributions

Y.K. and C.M. conceived the study and wrote the manuscript. Y.K. performed device preparation, measurement and analysis. J.K. synthesized polymers. J.K. and A.G. contributed to analyzing results. C.M. supervised the study.

## Funding

## Competing interests

The authors declare no competing interests.
