## [Peer Review File · Nature Communications]

Small Signal Analysis for the Characterization of Organic Electrochemical TransistorsREVIEWER COMMENTS

Reviewer #1 (Remarks to the Author):

Kim et al. describe in their manuscript “Small Signal Analysis for the Characterization of Organic Electrochemical transistors” a new protocol to reliably extract performance parameters of OECTs. As this topic is currently discussed intensively, this manuscript presents an interesting and timely contribution.

Overall, using small signal analysis to determine material parameters of OMIECS is highly interesting and relevant. However, we have some questions and suggestions for revision.

- Issues with current methods for calculating the mobility and volumetric capacitance are discussed in the introduction. Figure 1b plots the coefficient of variation. From which data is this plot extracted and are the numbers taken for different materials? How many devices were considered for each data point? Comparing data from different labs, different protocols and most likely different materials might be misleading if the authors compare it to their own data. We suggest to compare the coefficient of variation for their own data – extracted using the standard mechanism and the improved small signal method.

- μ is defined twice – once as mean (caption of Figure 1) and as mobility as well

- How is the onset voltage defined and how was it extracted?

- Overall, large sections of the manuscript excessively refer to the supplementary information only, which makes it challenging to follow the discussion. In particular the discussion of the ambipolar material relies on supplementary figures only. We suggest to either include the data in the main text or don't discuss the material in this manuscript.

- Unfortunately, the data obtained for PEDOT:PSS is discussed, but not included in the manuscript. As PEDOT:PSS is a standard material, we suggest to include it in the manuscript.

- Figure 3b displays a very strong dependency of the mobility on the gate voltage, which is surprising. How do you explain this strong dependence? Would you expect a saturation of mobility at larger gate voltages?

- You list the contact resistance as the cause of multiple phenomena observed during the calculations. You should explain this in more depth: Why is the peak in fig. 2g related to the contact resistance? How exactly can the contact resistance account for the apparently negative mobility?

- Starting from line 223, you start a discussion about the importance of considering parasitic effects when determining μ from small signal analysis since they contribute to ionic conduction. Discuss why and what effect they have on the mobility. How did you account for these effects?

- In Table 1, under “constant gate current”, C^* and μ are marked with 1) and 2). What do these references mean? Did you omit the caption of the table by accident?

- Typo: Line 256: decrease not decease

- The protocol used for the small signal characterization of the OECT is discussed starting line 384. At first, this section was confusing. Please note at the end of line 386 that a second channel is used to source the drain potential, different to the one used for the gate electrode.

Reviewer #2 (Remarks to the Author):

Reviewer #3 (Remarks to the Author):

Report NCOMMS-24-27404 “Small Signal Analysis for the Characterization of Organic Electrochemical Transistors” by Youngseok Kim et al.

Kim et al. present an analysis method that aims to resolve limitations of conventional methods to determine the device and material characteristics of organic electrochemical transistors (OECTs). The authors do so by means of a small signal AC analysis in the frequency domain combined with vector analysis. They claim that their method provides access to physical quantities such as conductance, electronic charge carrier mobility, and volumetric capacitance of significantly reduced uncertainty, while disentangling it from factors like the active layer thickness. The method is demonstrated for various material systems, covering the entire combinatorics of operating mode (depletion vs. accumulation) and active charge carrier type (n- vs. p-type). Further, it is benchmarked statistically and against established methods, showing convincing agreement. In summary, the method presented by Kim et al. is a novel approach to study OECTs and appears promising to overcome limitations of conventional methods. It seems suitable for Nature Comm. However, the manuscript needs to be revised to be ready for publication.

Major issues:

Page 2:

“The most important parameters are the volumetric capacitance C^* , representing the ability to accumulate/deplete charge carriers per unit volume through ionic conduction...”

This definition of capacitance is ambiguous. The intention is clear, but the sentence could benefit from a refinement in the interest of physical correctness.

Eq. (1)

The conditionals for the saturation and linear regime should be adjusted to involve absolute bars, in particular w.r.t. the fact that the paper deals with devices of both n- and p-type conductivity.

Page 3:

“In addition, the OECT active layer, which swells during operation due to the uptake of electrolyte, is typically only tens of nanometers thin.”

While it is true that the active layer swells due to electrolyte exposure, it is not clear that this process is due to the device operation (i.e., the supply and removal of ions) itself. Using an electrolyte like aqueous NaCl solution, swelling of the active layer will predominately be due to the uptake of water, rather than due to the insertion of the actual electrolyte ions Na and Cl. Nonetheless, the assessment that significant errors may arise from differences in film thicknesses between the swollen film in an OECT and the one measured by AFM or profilometry, is very reasonable.

Page 5:

“...while a constant = 10 mV was applied. The conjugated polymer p(g3TT-T2) with triethylene glycol side chains was used as the active material, ... which features ... outstanding on-current values of more than 100 μ A, an on-off current ratio of 10³ (Fig. S3a)...”

Presumably, the authors refer to , given that the active material is an electron hole conductor (this also applies to following pages). In that context, the authors name an on-current of >100 μ A, which is -dependent, and specifically seems to refer to instead of – 10 mV, as shown in Fig. S3a. Like this, the paragraph is misleading.

Page 6:

Fig.2

Fig. 2 aims to summarize the method the authors propose and therefore is central to the work. Yet, the figure would benefit from a generous revision in the interest of clarity. For instance, axis labeling should be consistent with what is written in the figure’s caption. The scale bar in the micrograph (Fig. 2a) is vacuous without a corresponding label. The phasor diagram (Fig. 2c) would benefit from an enlargement and perhaps further elaboration, given its central role to the method and following figures.

Minor issues:

Page 7:

“The electrochemical double layer at the interface between the electrolyte and active layer and the p-type operation of the material will give rise to coupling between ionic and electronic conduction.”

This statement is certainly not wrong, yet the authors might consider not only the double layer between OMIEC and electrolyte, but the actual microscopic coupling that arises between electrolyte ions and polymer strands.

Page 10:

Fig. 3

The plots show a very convincing agreement between the AC- and DC-methods. Do the authors have an explanation for the minimal, though consistent, offset to smaller values for their AC-method?

Point-by-point response to the reviewer comments.

Reviewer #1:

Kim et al. describe in their manuscript “Small Signal Analysis for the Characterization of Organic Electrochemical transistors” a new protocol to reliably extract performance parameters of OECTs. As this topic is currently discussed intensively, this manuscript presents an interesting and timely contribution. Overall, using small signal analysis to determine material parameters of OMIECS is highly interesting and relevant. However, we have some questions and suggestions for revision.

We are grateful for Reviewer #1 to appreciate the value of our research and give us constructive comments to improve the quality of our manuscript further. We did our best to reflect his/her comments in the revised manuscript as described in the following.

[Comment #1]

Issues with current methods for calculating the mobility and volumetric capacitance are discussed in the introduction. Figure 1b plots the coefficient of variation. (1) From which data is this plot extracted and are the numbers taken for different materials? (2) How many devices were considered for each data point? Comparing data from different labs, different protocols and most likely different materials might be misleading if the authors compare it to their own data. (3) We suggest to compare the coefficient of variation for their own data – extracted using the standard mechanism and the improved small signal method.

[Response #1]

We thank reviewer #1’s valuable comment on the plot of the mobility versus their coefficient of variation. (1, 2) Fig. 1b was collected from articles in which the mean and standard deviation values were described for materials, ranging from p-type PEDOT:PSS (depletion mode) and polythiophene-based polymers (accumulation mode), as well as n-type diketopyrrolopyrrole- and naphthalenediimide-based polymers (accumulation mode). In some cases, however, the reported number of samples was too small (usually 3 to 5 or not stated) without information on the method to obtain standard deviation values, so that we can’t conduct a more in-depth statistical analysis. (3) Further, most of the active materials depicted in Fig. 1 cannot be purchased, so that we can’t conduct any additional comparison with them. Please note that the references from which literature values were extracted are specified in the figure legend.

The complementary experiment with both conventional methods and suggested small signal analysis was carefully conducted with one identical organic mixed conductor p(g₃TT-T2), having high electrical mobility, volumetric capacitance and operational stability [35], as depicted in Fig. 4, and their results were included in Fig. 1 (blue diamond and red circle). We believe that Fig. 1 and 4 clearly show that the reliability of our suggested characterization is superior to conventional methods.

[Comment #2]

μ is defined twice – once as mean (caption of Figure 1) and as mobility as well.

[Response #2]

Following Reviewer #1's comment, the extracted mobility μ and the sample mean mobility $\bar{\mu}$ were defined with different notations in the revised manuscript.

At page 3,

“...An analysis of the recent literature indicates a coefficient of variation $\hat{c}_v = SD/\bar{\mu}$ of up to 100% for mobility values obtained via the traditional transfer curve/EIS method (standard deviation SD and mean mobility $\bar{\mu}$).”

At page 4,

Fig. 1. Correlation between the mean mobility $\bar{\mu}$ and the coefficient of variation \hat{c}_v . (a) Measurement scheme for the characterization of OECT devices. (b) Coefficient of variation $\hat{c}_v = SD/\mu$, where $\bar{\mu}$ and SD are the mean mobility and corresponding standard deviation, extracted from literature since 2014 (open symbols)^{6,9,10,19–34} and obtained in this work using the conventional transfer curve/EIS method (blue diamond) and small signal analysis (red circle); dashed lines represent $\hat{c}_v = 1, 5, 10$ and 100%.

[Comment #3]

How is the onset voltage defined and how was it extracted?

[Response #3]

The onset potential, which is the point where the measured/extracted values begin to rise, was obtained from the x-axis intercept of a linear fit near the maximum slope of the curve, which is the same way that is used to determine energy levels from cyclic voltammetry curves. To clearly explain this point, we add a description of the extraction method to the revised manuscript.

At Page 8,

“The change in C^* as a function of $V_{GS,DC}$ indicates that the active layer material is gradually doped for $V_{GS,DC} < -0.1$ V, which agrees with the onset potential extracted from the x-axis intercept of a linear fit near the maximum slope of the current transient of p(g3TT-T2) recorded during oxidation ($E_{onset} = +0.1$ V, vs. Ag/AgCl, see Fig. S2a).”

[Comment #4 and 5]

Overall, large sections of the manuscript excessively refer to the supplementary information only, which makes it challenging to follow the discussion. In particular the

discussion of the ambipolar material relies on supplementary figures only. (1) We suggest to either include the data in the main text or don't discuss the material in this manuscript.

Unfortunately, the data obtained for PEDOT:PSS is discussed, but not included in the manuscript. (2) As PEDOT:PSS is a standard material, we suggest to include it in the manuscript.

[Response #4 and 5]

(1) We thank Reviewer #1's constructive comment on the structure of the manuscript. In this manuscript, we suggest and emphasize a new device and material characterization method. We demonstrate the method for various OMIEC materials to show its general applicability and not with the purpose of discussing the details of each case.

Hence, as Reviewer #1 suggested, we moved the section that describes and discussed the measurement results of various materials to the supplementary information Note 1 to increase readability.

(2) Also, we apologize for not including the measurements of PEDOT:PSS devices, which was not our intention. The data for PEDOT:PSS are now included in Figure S9.

At page 13,

“To establish to which extent small signal analysis can be applied to a wide range of materials, we characterized OECT devices based on various OMIECs including p(g42T-T) (p-type, accumulation mode), PEDOT:PSS (p-type, depletion mode), and p(gNDI-gT2) (n-type/p-type, accumulation mode) (see SI Note 1 and Figs. S8-13 for detailed results of various OMIEC material based OECTs).”

In supplementary information,

SI Note 1. Small signal analysis of various OMIEC materials

All materials were characterized with a gate potential window of 1 V ($-0.6 \rightarrow +0.4$ V for p(g42T-T); $-0.2 \rightarrow +0.8$ V for PEDOT:PSS; $-0.8 \rightarrow +0.2$ V and $-0.5 \rightarrow +0.5$ V for p-/n-type operation of p(gNDI-gT2)) and a $V_{DS} = 0.01$ V for p(g42T-T) and PEDOT:PSS, and $V_{DS} = +0.1/-0.1$ V for p-/n-type operation of p(gNDI-gT2).

In case of OECTs based on p(g42T-T), I_{GS} and I_{DS} increased with $V_{GS,DC}$, showing an on-current value near 20 μ A and an on-off current ratio of 10^3 (Fig. S8a-c). All device characteristics are comparable to those of p(g3TT-T2) based OECTs (Fig. S8d-i). C^* increased for $V_{GS,DC} < 0.15$ V, revealing a shift in onset potential by 0.25 V compared to that of devices based on p(g3TT-T2), which is due to the lower ionization energy IE of the former ($IE = 4.3$ and 4.5 eV for p(g42T-T)¹ and p(g3TT-T2)²). μ_{AC} gradually increased as $V_{GS,DC} < V_{onset} - 0.35$ V, while a potential difference of -0.2 V is required in case of p(g3TT-T2) based OECTs (see Figs. 3c and 3f). In case of both materials, we explain the difference between the onset potentials of C^* and μ_{AC} with the structural changes that occur during oxidation, leading to the formation of a conducting network at which point the mobility increases.³ Also, the absence of a peak in $\Delta I'_{GS,AC}$ (data not shown), which was observed near the onset potential of C^* for p(g3TT-T2) (see Fig. 2g), indicates a lower contact resistance in case of p(g42T-T) based devices. OECTs based on

p(g42T-T) yield a maximum $[\mu C^*]_{AC} = 220 \text{ F cm}^{-1} \text{ V}^{-1} \text{ s}^{-1}$ a $\mu_{AC} = 1.25 \text{ cm}^2 \text{ V}^{-1} \text{ s}^{-1}$, which is in good agreement with previously reported values.^{4,5}

Fig. S8. Characterization of p(g42T-T) through small signal analysis. (a) Chemical structure of p(g42T-T). (b) I_{GS} (upper panel) and I_{DS} (lower panel) as a function of $V_{GS,DC}$. (c) $I_{DS,DC}$ and $I_{G,DC}$ as a function of $V_{GS,DC}$. Traces of (d) G (e) σ , (f) C^* , (g) g_m , (h) $[\mu C^*]_{AC}$, and (i) μ_{AC} as a function of $V_{GS,DC}$.

The depletion mode operation of PEDOT:PSS based OECTs (Fig. S9a) is evidenced by the device characteristics (see I_{GS} and I_{DS} vs. $V_{GS,DC}$ in Fig. S9b) showing an on-current of 60 μA at $V_{GS,DC} = 0 \text{ V}$, and a near zero value at $V_{GS,DC} > +0.7 \text{ V}$ with an on-off current ratio of 10^4 (Fig. S9c). $\Delta I_{GS,AC}$ and corresponding C^* values do not significantly change in the potential window from -0.2 to $+0.8 \text{ V}$, but decrease at $V_{GS,DC} > 0.8 \text{ V}$ (data not shown). Above 0.8 V , however, the device behavior becomes unstable, which we explain with permanent degradation due to excessive volume expansion as a result of cation uptake accompanied by the electrolyte solvent (water).⁶ Maximum values of $[\mu C^*]_{AC} = 351 \text{ F cm}^{-1} \text{ V}^{-1} \text{ s}^{-1}$ and $\mu_{AC} = 4.0 \text{ cm}^2 \text{ V}^{-1} \text{ s}^{-1}$ were achieved at $V_{GS,DC} = +0.2 \text{ V}$, in agreement with previous reports.⁷⁻¹⁰

Fig. S9. Characterization of PEDOT:PSS through small signal analysis. (a) Chemical structure of PEDOT and PSS. (b) I_{GS} (upper panel) and I_{DS} (lower panel) as a function of $V_{GS,DC}$. (c) $I_{DS,DC}$ and $I_{G,DC}$ as a function of $V_{GS,DC}$. Traces of (d) G (e) σ , (f) C^* , (g) g_m , (h) $[\mu C^*]_{AC}$, and (i) μ_{AC} as a function of $V_{GS,DC}$.

Prior to small signal analysis of OECTs based on the ambipolar material p(gNDI-gT2), we conducted EIS, cyclic voltammetry (CV) and conventional OECT characterization because of the complexity of mixed p-type and n-type operation. Cyclic voltammograms (Fig. S10a) confirm that the material shows a combination of both types of electrochemical response at $E > +0.5$ V and $E < -0.15$ V, with two distinct volumetric capacitance values of $C^* = 150$ F cm⁻³ at $E = +0.8$ V and $C^* = 225$ F cm⁻³ at $E = -0.5$ V (Fig. S10b-d), which are associated with hole-anion and electron-cation accumulation, respectively. The output characteristics of p-type operation devices (Fig. S11a) exhibit clear linear and saturation regimes, which are evident for $V_{GS} < -0.4$ V, while n-type operation is observed for $V_{GS} > -0.4$ V and $V_{DS} < -0.3$ V (see the onset potential during reduction in Fig. S10a).¹¹ Instead, output characteristics for n-type operation also show linear and saturation mode operation at $V_{GS} > -0.2$ V (Fig. S11b), resulting

in ambipolar behavior (Fig. S11c and d).

Fig. S10. Electrochemical characterization of p(gNDI-gT2). (a) Cyclic voltammetry of thin films for 5 cycles. Impedance Z (upper panel) and phase curves (lower panel) as a function of the offset potential E vs. Ag/AgCl (b) from 0.0 (black) to +0.8 V (red), and (c) from 0.0 (black) to -0.5 V (blue). (d) Extracted volumetric capacitance C^* as a function of E .

Fig. S11. (a-b) Output and (c-d) transfer curves of p(gNDI-gT2)-based OECTs, which were obtained by the conventional characterization method for (a, c) p-type and (b, d) n-type operation. In the transfer curves, the electronic conduction by opposite charge carriers is remarked with a gray color.

In case of p-type operation, both I_{GS} and I_{DS} gradual increase as $V_{GS,DC}$ changes from -0.4 to -0.6 V (see Fig. S12a-b). In contrast to the transfer curves recorded in the saturation regime (see Fig. S11c), only the electronic current by holes is monitored at $V_{GS,DC} < 0$, attributed to the low amplitude of V_{DS} of $+0.1$ V (Fig. S12c). Further, $I_{GS,DC}$ and $I_{DS,DC}$ agree with a transfer curve recorded using the conventional method. Values of $G = 3.7 \mu\text{S}$ and $\sigma = 135 \text{ mS cm}^{-1}$ are lower than those reported for other p-type materials, despite of a high $C^* = 125 \text{ F cm}^{-3}$ at $V_{GS,DC} = -0.8$ V. Maximum g_m and $[\mu C^*]_{AC}$ values of $2.27 \mu\text{S}$ and $0.84 \text{ F cm}^{-1} \text{ V}^{-1} \text{ s}^{-1}$ are achieved at $V_{GS,DC} = -0.7$ V, while the mobility reaches a value of $\mu_{AC} = 0.0076 \text{ cm}^2 \text{ V}^{-1} \text{ s}^{-1}$ at $V_{GS,DC} = -0.7$ V.

Fig. S12. Characterization of p(gNDI-gT2) in p-type operation through small signal analysis. (a) Chemical structure of p(gNDI-gT2). (b) I_{GS} (upper panel) and I_{DS} (lower panel) as a function of $V_{GS,DC}$. (c) $I_{DS,DC}$ and $I_{G,DC}$ as a function of $V_{GS,DC}$. Traces of (d) G (e) σ , (f) C^* , (g) g_m , (h) $[\mu C^*]_{AC}$, and (i) μ_{AC} as a function of $V_{GS,DC}$.

In case of n-type operation, the electrochemical and electrical response occurred for positive $V_{GS,DC}$ ($+0.05$ and $+0.15$ V, respectively) (Fig. S13a-c). The negative slope of I_{DS} above $V_{GS,DC} = +0.36$ V resulted in a negative slope in G and σ traces (Fig. S13d-e). Even though, C^* values

gradually increased to 200 F cm^{-3} (Fig. S13f), negative g_m , $[\mu C^*]$, and μ values were obtained (Fig. S13g-h), which are the main characteristics of anti-ambipolar transistors that can be employed for the construction of neuromorphic devices,¹² frequency doubler circuits¹³ and ternary logic circuits.¹⁴ Considering the inversion of I_{DS} above $V_{GS,DC} = -0.4 \text{ V}$, as shown in the output characteristic curves (see Fig. S11b), we attribute the negative values of g_m , $[\mu C^*]$, and μ to increased contact resistance at $V_{GS,DC} > 0.36 \text{ V}$.¹⁵ Note that the apparent negative μ value does not imply opposite direction of charge-carrier motion (Fig. S13i), but may arise due to the parasitic contact resistance, opening of a hard Coulomb gap,¹⁶ and/or band-to-band tunneling behavior¹⁷ etc.

Fig. S13. Characterization of p(gNDI-gT2) in n-type operation through small signal analysis. (a) Chemical structure of p(gNDI-gT2). (b) I_{GS} (upper panel) and I_{DS} (lower panel) as a function of $V_{GS,DC}$. (c) $I_{DS,DC}$ and $I_{G,DC}$ as a function of $V_{GS,DC}$. Traces of (d) G (e) σ , (f) C^* , (g) g_m , (h) $[\mu C^*]_{AC}$, and (i) μ_{AC} as a function of $V_{GS,DC}$.

[Comment #6]

Figure 3b displays a very strong dependency of the mobility on the gate voltage, which is surprising. (1) How do you explain this strong dependence? (2) Would you expect a saturation of mobility at larger gate voltages?

[Response #6]

(1) Voltage-dependent mobility (or transconductance) is a common phenomenon in disordered organic semiconducting materials. The charge carriers should transport through the high energy barriers between locally-doped polymer chains via a hopping process, and the density of doped polymer chains can be varied by the applied gate potential. It means that the charge carriers face lower energy barriers as the number of the doped chains is increased, so that the carrier mobility tends to be increased at higher doping levels.^{R1-R2} Further, these mobility values decrease again when the material is heavily doped in case of the organic mixed conductors. The reason is not clearly explained yet, but possible reasons are i) farther distance between disordered polymer chains due to excessive swelling^{R3} and 2) increased presence of less-mobile bipolarons at high doping levels.^{R4-R5} Since transport in the organic mixed ionic–electronic conductors typically involves a voltage-dependent mobility, only the maximum mobility and the maximum capacitance values are considered to determine the material- and device-performance.^{R6} This is also why the analysis of overall responses over the entire V_g window is important for in-depth understanding of coupled ionic-electronic conduction, as we demonstrated in the manuscript.

(2) In case of the material p(g₃TT-T2) which was mainly used in the manuscript, the mobility also does not show a plateau, but decreased at $V_{GS} < -0.6$ V.³⁶

- R1. Coehoorn, R., Pasveer, W. F., Bobbert, P. A. & Michels, M. A. J. Charge-carrier concentration dependence of the hopping mobility in organic materials with Gaussian disorder. *Phys. Rev. B* 72, 155206 (2005).
- R2. Arkhipov, V. I., Emelianova, E. V., Heremans, P. & Bäessler, H. Analytic model of carrier mobility in doped disordered organic semiconductors. *Phys. Rev. B* 72, 235202 (2005).
- R3. Paulsen, B. D. & Frisbie, C. D. Dependence of Conductivity on Charge Density and Electrochemical Potential in Polymer Semiconductors Gated with Ionic Liquids. *J. Phys. Chem. C* 116, 3132–3141 (2012).
- R4. Enokida, I. & Furukawa, Y. Doping-level dependent mobilities of positive polarons and bipolarons in poly(2,5-bis(3-hexadecylthiophen-2-yl)thieno[3,2-b]thiophene) (PBTTC16) based on an ionic-liquid-gated transistor configuration. *Org. Electron.* 68, 28–34 (2019).
- R5. Voss, M. G. et al. Driving Force and Optical Signatures of Bipolaron Formation in Chemically Doped Conjugated Polymers. *Adv. Mater.* 33, 2000228 (2021).
- R6. Inal, S., Malliaras, G. G. & Rivnay, J. Benchmarking organic mixed conductors for transistors. *Nat. Commun.* 8, 1767 (2017).

As a response to Reviewer #1's comment, we add a description of voltage-dependent mobility of p(g₃TT-T2) to the revised manuscript.

At page 9,

“...For p(g₃TT-T2), μ_{AC} gradually increased with increasing doping level for $V_{GS,DC} < -0.3$ V (solid line, Fig. 3f), and a highest value of $1.95 \text{ cm}^2 \text{ V}^{-1} \text{ s}^{-1}$ was achieved at $V_{GS,DC} = -0.6$ V. At $V_{GS,DC} < -0.6$ V, μ_{AC} tends to decrease slightly (data not shown), which is expected due to formation of less-mobile bipolarons^{39,40} and/or an increased distance between polymer chains due to excessive swelling.⁴¹...”

[Comment #7]

You list the contact resistance as the cause of multiple phenomena observed during the calculations. You should explain this in more depth: (1) Why is the peak in fig. 2g related to the contact resistance? (2) How exactly can the contact resistance account for the apparently negative mobility?

[Response #7]

(1) In the parameter extraction from the electrochemical impedance spectroscopy, we used the equivalent circuit model $R_1[R_m C_m (R_{contact} [R_{active} C_{active}])]$, where R_1 is a resistance in lead wires and the electrolyte, R_m and C_m are a resistive and capacitive impedance at the surface of metal electrodes, R_{active} and C_{active} are a resistive and capacitive impedance of the active layer, and $R_{contact}$ is the contact resistance between the metal electrode and OMIEC layer, as depicted in Fig. S2c. Further, through the equivalent circuit fitting, we can extract $R_{contact}$ as a function of the potential E in the same manner as for the volumetric capacitance. Further, the resistivity $\rho_{contact}$ can be calculated by dividing $R_{contact}$ with the contact area. As depicted in Fig. R1, $\rho_{contact}$ shows very low values below $10 \text{ } \Omega \text{ cm}^2$ when the material is doped at $E > 0.2$ V, but gradually increased up to $620 \text{ } \Omega \text{ cm}^2$ and tends to reach a plateau as the active material becomes dedoped at $E < 0.2$ V. At $E < 0$ V, note that the current flow through the active material is too low due to the high impedance of the active material (*i.e.*, low capacitance at $E < 0$ V as shown in Fig. S2b), so that the extracted $\rho_{contact}$ values at $E < 0$ V seems to have high deviations with a low accuracy.

Fig. R1. Extracted contact resistivity $\rho_{contact}$ as a function of E .

In the case of small signal analysis, all the parameters were extracted at a frequency of 10 Hz, so that only one resistive and one capacitive component can be recognizable when analyzing the real and imaginary component, respectively. Further, once the additional resistive component (*i.e.*, contact resistance) is included during the measurement, it is added on the resistive component, thus resulting in the resistive current response at the gate electrode.

In the revised manuscript, the data from Fig. R1 were added to Fig. S2, and explained in the main text.

At page 8,

“...This peak is assigned to the contact resistance between the active layer and metal electrode (Fig. S2d), which is also observed in corresponding EIS measurements (Fig. S4; *c.f.* the resistive response showing a flat region near f_{AC} 10 Hz in the Bode plot). Unlike the EIS measurement, the small signal analysis is conducted with one frequency value, so that the parasitic response is superimposed onto the current value.”

In supplementary information,

Fig. S2. Electrochemical characterization of p(g₃TT-T₂). (a) Cyclic voltammetry of thin films for 5 cycles. (b) Impedance Z (upper panel) and phase curves (lower panel) as a function of the offset potential E vs. Ag/AgCl from -0.4 to +0.6 V. (c) Equivalent circuit for the circuit fitting, and (d) extracted volumetric capacitance C^* and contact resistivity $\rho_{contact} = R_{contact}/\text{contact area}$ as a function of E .

(2) We thank Reviewer #1 for pointing out our weak argument about the negative mobility of an ambipolar OMIEC material. The output characteristic curve of p(gNDI-gT2) (Fig. S11b) clearly depicts the non-linear drain current response at the low V_{DS} and in case of $V_{GS} > 0.5$ V, indicating the presence of non-ohmic contact resistance. However, there are other reasons for the negative mobility, and we cannot provide additional experimental evidence. So, we refined our argument in a revised manuscript.

In supplementary information,

“...Note that the apparent negative μ value does not imply opposite direction of charge-carrier motion (Fig. S13i), but may arise due to the parasitic contact resistance, opening of a hard Coulomb gap,¹⁶ and/or band-to-band tunneling behavior¹⁷ etc..”

[Comment #8]

Starting from line 223, you start a discussion about the importance of considering parasitic effects when determining μ from small signal analysis since they contribute to ionic conduction. Discuss why and what effect they have on the mobility. How did you account for these effects?

[Response #8]

The parasitic resistive and capacitive components do not affect the actual mobility, but affect extraction of the parameter because all the equations written in the manuscript are only valid in case of ideal device operation. For instance, the small metallic gate electrode with two-electrode configuration definitely has a high impedance in the ionic conduction path, thus resulting in an undesired potential drop at the gate electrode, which reduces the actual electrochemical potential at the active material.^{R7} Further, all the resistive components (e.g. resistance at the contact and lead wires) except the channel resistance in electronic conduction path also distribute the applied drain potential, thus resulting in a lower electric field at the channel.^{R8-R9}

To minimize the possible confusion, we refined the paragraph and cite the articles in the revised manuscript.

R7. Tarabella, G. et al. Effect of the gate electrode on the response of organic electrochemical transistors. *Appl. Phys. Lett.* 97, 123304 (2010).

R8. Donahue, M. J. et al. High-Performance Vertical Organic Electrochemical Transistors. *Adv. Mater.* 30, 1705031 (2018).

R9. Kaphle, V., Liu, S., Al-Shadeedi, A., Keum, C.-M. & Lüssem, B. Contact Resistance Effects in Highly Doped Organic Electrochemical Transistors. *Adv. Mater.* 28, 8766–8770 (2016).

Page 12,

“...Furthermore, it is necessary to consider additional capacitive or resistive components that contribute to ionic conduction such as (1) a parasitic capacitance at the gate electrode (especially in case of a two-electrode configuration with a small gate electrode consisting of a noble metal or OMIEC such as PEDOT:PSS)⁴³ and/or at the metal source/drain electrode (e.g., electrical double layer capacitance), or (2) a parasitic resistance at the lead wires and interface between the active layer and source/drain metal electrode.^{44,45} Those parasitic components can lead to an incorrect gate potential at the channel, an overestimated C^* , and an incorrect drain potential through the channel, respectively.”

[Comment #9]

In Table 1, under “constant gate current”, C^* and μ are marked with 1) and 2). What do these references mean? Did you omit the caption of the table by accident?

[Response #9]

Thank you for pointing out these mistakes in our manuscript. We removed the superscript on Table 1 in the revised manuscript.

Table 1. Input/output signals and extractable parameters G , σ , C^* , g_m , $[\mu C^*]$ and μ for conventional OECT characterization methods (transfer curve/EIS, constant gate current, sinusoidal gate potential) and small signal analysis.

	transfer curve	EIS	constant gate current	sinusoidal gate potential	small signal analysis
input signal	V_{GS}, V_{DS}	stepwise V_{DC} + V_{AC}	stepwise I_{GS}, V_{DS}	stepwise V_{DC} + $V_{GS,AC}, V_{DS}$	continuous $V_{GS,DC}$ + $V_{GS,AC}, V_{DS}$
output signal	I_{GS}, I_{DS}	I_{AC}	V_{GS}, I_{DS}	$I_{GS,AC}, I_{DS,AC}$	$I_{GS,DC}, I_{GS,AC}, I_{DS,DC}, I_{DS,AC}$
G, σ, g_m	✓	-	✓	✓	✓
$[\mu C^*]$	✓	-	✓	✓	✓
C^*	-	✓	✓	✓	✓
μ	(✓)	-	✓	✓	✓

[Comment #10]

Typo: Line 256: decrease not decease

[Response #10]

As Reviewer #1 pointed out, we have corrected the typo in the manuscript.

In supplementary information,

“... $\Delta I_{GS,AC}$ and corresponding C^* values do not significantly change in the potential window from -0.2 to $+0.8$ V, but decrease at $V_{GS,DC} > 0.8$ V (data not shown). ...”

[Comment #11]

The protocol used for the small signal characterization of the OECT is discussed starting line 384. At first, this section was confusing. Please note at the end of line 386 that a second channel is used to source the drain potential, different to the one used for the gate electrode.

[Response #11]

We refined the method section for readers to demonstrate the small signal analysis with the easiest way.

At page 21,

Small signal analysis was conducted with a two-channel electrochemical workstation (Biologic, SP-300). Both channels for the gate and drain potential were set to the ‘CE to ground’ electrode connection mode available through the EC-Lab software, to assign the voltage based on the potential of the source electrode. The former and latter electrode were operated with a three-electrode and two-electrode configuration, respectively. For application of the gate potential, the Ag/AgCl electrode and Pt electrode were connected to the S1 and P1 ports respectively, and the source electrode of the device was connected to the S2, S3 and GND ports (S1: high potential, P1: high current, S2 and S3: low potential, GND: ground). For application of the drain potential, the drain electrode was connected to the P1 and S1 ports, and the source electrode was connected to the S2, S3 and GND electrode ports of the instrument. The gate and drain potentials were assigned with the ‘AC-Voltammetry’ and ‘Constant-Voltage’ functions available through the EC-Lab software, respectively (AC-Voltammetry; triangular potential with superimposed small sinusoidal waveform, Constant-Voltage; constant voltage bias, Waveform parameters are described in the manuscript). The specific potential values for each type of device are described in the manuscript. For a synchronized measurement on both channels, the built-in ‘synchronize’ function was used.

Reviewer: 3

Kim et al. present an analysis method that aims to resolve limitations of conventional methods to determine the device and material characteristics of organic electrochemical transistors (OECTs). The authors do so by means of a small signal AC analysis in the frequency domain combined with vector analysis. They claim that their method provides access to physical quantities such as conductance, electronic charge carrier mobility, and volumetric capacitance of significantly reduced uncertainty, while disentangling it from factors like the active layer thickness. The method is demonstrated for various material systems, covering the entire combinatorics of operating mode (depletion vs. accumulation) and active charge carrier type (n- vs. p-type). Further, it is benchmarked statistically and against established methods, showing convincing agreement. In summary, the method presented by Kim et al. is a novel approach to study OECTs and appears promising to overcome limitations of conventional methods. It seems suitable for Nature Comm. However, the manuscript needs to be revised to be ready for publication.

We thank Reviewer #3 for his/her positive comments on our research. Below, we responded Reviewer #3's comments, and manuscript was also fully reflected on the comments.

[Comment #1]

“The most important parameters are the volumetric capacitance C^* , representing the ability to accumulate/deplete charge carriers per unit volume through ionic conduction...” This definition of capacitance is ambiguous. The intention is clear, but the sentence could benefit from a refinement in the interest of physical correctness.

[Response #1]

We thank Reviewer #3 for the comment on the definition of the volumetric capacitance. The sentence to explain the volumetric capacitance is corrected in the revised manuscript.

At page 2,

“The most important parameters are the volumetric capacitance C^* , representing the change of the number of charge carriers stored per unit volume upon a small fluctuation in potential, and the mobility μ , *i.e.* the potential-normalized velocity of electronic charge carriers.¹³”

[Comment #2]

Eq. (1); The conditionals for the saturation and linear regime should be adjusted to involve absolute bars, in particular w.r.t. the fact that the paper deals with devices of both n- and p-type conductivity.

[Response #2]

As recommended by Reviewer #3, we have stated the equation in general form.

At page 2,

A widely used method for the determination of C^* and μ involves two measurements: (1) OEET characterization and the analysis of transfer curves and (2) electrochemical impedance spectroscopy (EIS).¹³ OEET measurements allow to determine the volumetric transconductance g_m^* from transfer curves, *i.e.* the volumetric source-drain current I_{DS}^* is recorded as a function of gate potential V_{GS} , according to:

$$g_m^* = g_m / \frac{wd}{L_{ch}} = \frac{dI_{DS}^*}{dV_{GS}} = \begin{cases} [\mu C^*] \cdot V_{DS}, & \text{linear regime} \\ [\mu C^*] \cdot (V_{GS} - V_{th}), & \text{saturation regime} \end{cases} \quad (1)$$

where g_m is the transconductance, w , d and L_{ch} are the width, thickness, and length of the channel, respectively, V_{DS} is the drain potential and V_{th} is the threshold voltage. The product $[\mu C^*]$ can thus be obtained from transfer curves in either the linear ($V_{DS} > V_{GS} - V_{th}$ for p-type and $V_{DS} < V_{GS} - V_{th}$ for n-type OEETs) or saturation regime ($V_{DS} < V_{GS} - V_{th}$ for p-type and $V_{DS} > V_{GS} - V_{th}$ for n-type OEETs).

[Comment #3]

“In addition, the OEET active layer, which swells during operation due to the uptake of electrolyte, is typically only tens of nanometers thin.”

While it is true that the active layer swells due to electrolyte exposure, it is not clear that this process is due to the device operation (*i.e.*, the supply and removal of ions) itself. Using an electrolyte like aqueous NaCl solution, swelling of the active layer will predominately be due to the uptake of water, rather than due to the insertion of the actual electrolyte ions Na and Cl. Nonetheless, the assessment that significant errors may arise from differences in film thicknesses between the swollen film in an OEET and the one measured by AFM or profilometry, is very reasonable.

[Response #3]

As an OMIEC material is doped/dedoped, not only the ions but also the surrounded water molecules are both transported into the polymer network due to favorable interactions between ions and polar water molecules. The paper written by Camila Cendra *et. al.* clearly demonstrated through electrochemical quartz crystal microbalance with dissipation monitoring that the swelling ratio is increased as the size of ions is smaller, which can bring more water ions.^{R10}

To minimize the possible confusion, we described it in detail, and cited the abovementioned paper.

R10. Cendra, C. *et al.* Role of the Anion on the Transport and Structure of Organic Mixed Conductors. *Adv. Funct. Mater.* 29, 1807034 (2019).

At page 3,

“...The use of two distinct characterization methods for the determination of $[\mu C^*]$ and C^* introduces uncertainty in the calculated mobility due to the propagation of errors. In addition, the OEET active layer, which swells during operation due to the uptake of hydrated ions (*i.e.*, ions that are accompanied by water molecules¹⁵), is typically only tens of nanometers thin. As

CHALMERS

a result, a large relative error in d can arise due to an uneven film thickness across the channel, the adventitious uptake of water, as well as errors inherent to film thickness measurements with atomic force microscopy (AFM) or surface profilometry...”

[Comment #4]

Page 5: “...while a constant = 10 mV was applied. The conjugated polymer p(g₃TT-T2) with triethylene glycol side chains was used as the active material, ... which features ... outstanding on-current values of more than 100 μ A, an on-off current ratio of 103 (Fig. S3a)...”

Presumably, the authors refer to τ , given that the active material is an electron hole conductor (this also applies to following pages). In that context, the authors name an on-current of $>100 \mu$ A, which is τ -dependent, and specifically seems to refer to τ instead of -10 mV, as shown in Fig. S3a. Like this, the paragraph is misleading.

[Response #4]

Authors thank Reviewer #3 for valuable comments. To prevent misunderstanding about properties of the active material, we corrected the sentence with minimized information as written below.

At page 5,

“...The conjugated polymer p(g₃TT-T2) with triethylene glycol side chains was used as the active material (see **Fig. S1** for chemical structure), which features a high electronic mobility upon electrochemical oxidation (**Figs. S2 and S3**) with good stability during cyclic operation.³⁶”

[Comment #5]

Fig. 2 aims to summarize the method the authors propose and therefore is central to the work. Yet, the figure would benefit from a generous revision in the interest of clarity. For instance, axis labeling should be consistent with what is written in the figure’s caption. The scale bar in the micrograph (Fig. 2a) is vacuous without a corresponding label. The phasor diagram (Fig. 2c) would benefit from an enlargement and perhaps further elaboration, given its central role to the method and following figures.

[Response #5]

Authors thank Reviewer #3 for valuable comment on the Fig. 2. As recommended, we edited the figure and figure caption in the revised manuscript.

Fig. 2

Fig. 2. Small signal analysis for OECT device characterization. (a) Device and measurement scheme for small signal analysis. A gate potential V_{GS} consisting of a triangular potential $V_{GS,DC}$ and a small-amplitude sinusoidal potential $V_{GS,AC}$ is applied to the 100 mM NaCl aqueous electrolyte via a three-electrode configuration with a counter (CE) and a working electrodes (WE), with a constant drain potential V_{DS} . The lower inset depicts an optical microscopy image of the channel region of an OECT device (scale bar = 100 μm). (b) Time-varying input gate potential V_{GS} (black) as well as output gate I_{GS} (red) and drain current I_{DS} (blue). (c) Phasor diagram of output currents relative to V_{GS} , and (d) traces of V_{GS} (black), I_{GS} (red) and I_{DS} (blue) at $t = 95 \text{ s} + \Delta t$, and centered at $V_{GS} = 0.55 \text{ V}$, $I_{GS} = 0 \text{ A}$, and $I_{DS} = 8 \mu\text{A}$ (y-axis scale bar = 10 mV, 100 nA, and 1 μA for V_{GS} , I_{GS} , and I_{DS}). Since the holes are accumulated through the electrical double layer in case of p-type accumulation mode OECTs (see inset of (c)), the gate and drain currents show a time delay from the gate input bias of $\pi/2$ and π , respectively. (e) Amplitude of AC and DC components of I_{GS} (upper panel) and I_{DS} (lower panel) as a function of $V_{GS,DC}$ (f) Steady-state current response $I_{DS,DC}$ (solid line) and $I_{GS,DC}$ (dashed line) as a function of the $V_{GS,DC}$ from small signal analysis, and transfer curve I_{DS} vs. $V_{GS,DC}$ from conventional analysis (open circles). (g) Amplitude of $I_{GS,AC}$ (upper panel) and $I_{DS,AC}$ (lower panel) and its real and imaginary components as a function of $V_{GS,DC}$, obtained from small signal analysis. The real (red) and imaginary component (blue) were extracted from the output current values (black) by vector analysis as shown in (c).

[Comment #6]

Page 7: “The electrochemical double layer at the interface between the electrolyte and active layer and the p-type operation of the material will give rise to coupling between ionic and electronic conduction.” This statement is certainly not wrong, yet the authors might consider not only the double layer between OMIEC and electrolyte, but the actual microscopic coupling that arises between electrolyte ions and polymer strands.

[Response #6]

We thank for Reviwer #3’s valuable comment. In the sentence, we intended to explain why the gate and drain currents have a phase shift of -90° and -180° relative to the gate potential. The sentence was refined in the revised manuscript to minimize confusion. Further, the microscopic ionic-electronic coupling in the molecular level is quite specific and not of interest in this manuscript, so it is not described in this manuscript.

At page 7,

“...Considering the electrochemical double layer at the interface between the electrolyte and active layer³⁶ and the p-type operation of the material,³⁵ a sinusoidal current response at the gate and drain electrode, $I_{GS,AC}$ and $I_{DS,AC}$ is expected with a phase shift of -90° and -180° relative to the $V_{GS,AC}$ signal, respectively.”

[Comment #7]

Fig. 3; The plots show a very convincing agreement between the AC- and DC-methods. Do the authors have an explanation for the minimal, though consistent, offset to smaller values for their AC-method?

[Response #7]

We expect that the smaller values of g_m and $[\mu C^*]$ arise due to the underestimated capacitance. In case of the mobility obtained from the DC response (*i.e.*, $\mu_{DC} = [\mu C^*] / C^*$), the value is overestimated because $[\mu C^*]$, which contains the number of charge carriers transferred through the channel, is divided by C^* , which contains the underestimated number of charge carriers through the electrolyte at 10 Hz operation frequency.

In the revised manuscript, we addressed the reason for small differences between the obtained values from AC and DC method.

At page 9,

“...Since small signal analysis underestimates C^* , values for $[\mu C^*]_{AC}$ obtained from the AC response are about 7% smaller than $[\mu C^*]_{DC}$ from the DC response ...”

REVIEWERS' COMMENTS

Reviewer #1 (Remarks to the Author):

The authors have satisfactorily addressed our comments and we recommend publication.

Reviewer #2 (Remarks to the Author):

Reviewer #3 (Remarks to the Author):

The authors have answered all questions with their extensive reply and new data and from my point of view the paper is ready for publication.

Reviewer #4 (Remarks to the Author):
